# TEST-TIME ZERO-SHOT RECOGNITION WITH GOOD ATTRIBUTES

## ABSTRACT

Test-time adaptation (TTA) has emerged as a zero-shot learning approach to address distribution shifts across domains without needing source data. While current methods focus on adapting vision and language models (VLMs) using prompt tuning, they struggle with ambiguous categories due to the challenge of selecting relevant attributes in the absence of labels. To address this issue, we propose a novel framework, termed Search4Prompt, which aims to identify "good" attributes and learn tailored prompts during test-time prompt learning (TTPL). Search4Prompt consists of two main components: the Retrieval-based Attribute Search (RAS) and the Implicit-Explicit Attribute Injection (IEAI) module. RAS constructs an attribute bank by generating detailed descriptions for predefined categories, and then identifies the most relevant attributes based on the semantic similarity between the test image and the attributes. This enables the selection of "good" attributes that are well-suited to the test samples. The IEAI module operates in two ways. First, it employs pseudo-label learning, where the selected attributes contribute to a voting process that implicitly injects attribute knowledge into prompt learning. Second, it augments the original category names with the selected attributes, explicitly enhancing the semantic representation of ambiguous categories. This dual approach improves the model's discriminability during test-time prompt learning. Experimental results demonstrate that Search4Prompt outperforms existing TTA methods on several benchmark datasets, confirming its effectiveness in narrowing domain gaps and handling ambiguous categories.

## 1 INTRODUCTION

Vision-language foundation models like CLIP (Radford et al., 2021) have demonstrated remarkable generalization across various recognition tasks. These models are pre-trained on large-scale web data using a contrastive loss and can be fine-tuned for specific tasks with additional training data. While this approach has led to significant advancements, its success is highly dependent on the assumption that there is no distribution shift between the training and test data. In reality, distribution shifts are common due to factors such as natural variations or changes in sensing equipment, making it impractical to collect sufficient training data for every possible test domain. To address these challenges, test-time adaptation (TTA) has emerged as a strategy to adapt pre-trained models to test data by minimizing entropy, without the need for source data (Kim et al., 2020; Shanmugam et al., 2021; Wang et al., 2020; Niu et al., 2023).

Among TTA methods (Wang et al., 2020; 2022; Yang et al., 2023a), test-time prompt learning (TTPL) (Shu et al., 2022; Abdul Samadh et al., 2024) has gained prominence for maintaining the generalization capabilities of vision-language models (VLMs) (Kumar et al., 2022). Inspired by natural language processing (NLP), TTPL introduces learnable prompt vectors by appending them to predefined class names. However, the performance of these models deteriorates when faced with ambiguous categories, particularly in tasks requiring fine-grained recognition. As illustrated in the "baseline" of Figure 1, broad or overlapping categories such as "red velvet cake", "cupcakes", and "carrot cake" are not adequately represented by the learned prompt vectors, leading to reduced effectiveness.

A natural approach to address ambiguous categories is to expand their definitions with detailed attributes. One potential solution, though unexplored in TTPL, is to leverage large language models

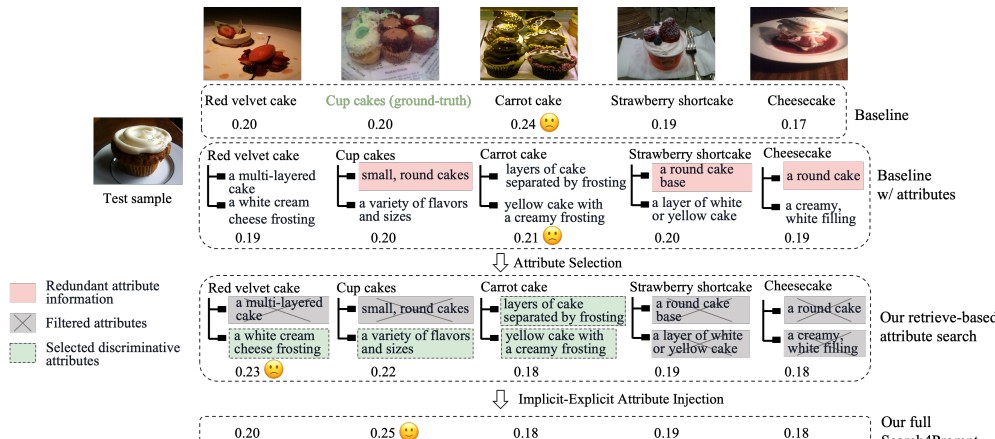

Figure 1: When dealing with visually similar objects, coarse-grained categories often create ambiguities that hinder accurate recognition. Ambiguous categories, however, can be clearly distinguished using specific fine-grained attributes. Yet, too many attributes (e.g., "round cakes") can introduce redundancy and confuse the model. Our method selectively identifies "good" discriminative attributes (e.g., flavor or size variations) from a broad set, ensuring precise visual recognition by focusing on the most relevant and informative characteristics.

(LLMs) to generate fine-grained attribute descriptions based on predefined category names and use these attributes to enhance TTPL. However, as illustrated in Figure 1, LLM-generated descriptions, while rooted in common sense, may not accurately correspond to specific samples in downstream tasks. Furthermore, an overabundance of attributes can introduce redundant information, potentially leading to prediction errors. This issue is particularly problematic in test-time adaptation scenarios, where only test samples are available and no labels are provided to guide selecting relevant, informative attributes.

To address these challenges, we propose a novel framework called Search4Prompt, designed to identify representative "good" attributes and learn tailored prompts for test samples during TTPL. The framework consists of two main components: the Retrieval-based Attribute Search (RAS) and the Implicit-Explicit Attribute Injection (IEAI) module. RAS begins by generating extensive attribute descriptions for predefined categories, creating an attribute bank. It then identifies the most relevant attributes by comparing the similarities between the CLIP features of the test image and the attributes, allowing us to find "good" discriminative attributes that are best suited for each sample.

To fully leverage the identified attributes, IEAI injects this attribute knowledge into TTPL through both implicit and explicit means. Inspired by how humans recognize objects by focusing on key details, we introduce a voting-based pseudo-label learning method, where the selected attributes are combined into scores that act as soft pseudo-labels. By aligning model predictions with these soft labels, we implicitly guide VLMs to focus on the most important features, especially for ambiguous or fine-grained categories.

Additionally, the success of adapting VLMs to downstream tasks often depends on the quality of category descriptions, which are typically fixed during TTPL (Shu et al., 2022; Abdul Samadh et al., 2024). To improve alignment with test samples, we implement a compositional attribute augmentation strategy, which enriches category descriptions with the most relevant attributes. This strategy detects ambiguous categories using pre-trained VLMs and augments them with the searched attributes, providing richer and more informative context. By improving the quality of category descriptions, our method ensures more effective generalization to test data, even in challenging scenarios.

In summary, our contributions are fourfold:

- We propose a novel Search4Prompt framework that discovers discriminative attributes from rich contextual information for learning more effective, informative prompts during test-time adaptation.
- We introduce the Retrieval-based Attribute Search module to effectively identify the most relevant attributes that are important for each testing sample.
- We present the Implicit-Explicit Attribute Injection module to facilitate the pseudo-labeling and prompt learning with the selected attributes.

- Experimental results demonstrate that our approach outperforms existing methods across extensive benchmark datasets. For instance, on the Flower dataset, our method achieves an improvement of 2.05% in accuracy compared to state-of-the-art methods.

## 2 RELATED WORK

**Prompt Learning in Vision-Language Models.** Vision-language models (VLMs) (Radford et al., 2021; Jia et al., 2021), pre-trained on vast image-text pairs from the web, have demonstrated powerful zero-shot generalization across downstream recognition tasks. However, efficiently adapting VLMs to specific downstream tasks with limited data remains a challenge. Recently, prompt learning has become a leading approach for adapting VLMs with unlabeled (Shu et al., 2022; Feng et al., 2023; Ma et al., 2024) or limited labeled data.

CoOp used the fine-tuning of VLMs by optimizing continuous prompt vectors in the language branch for few-shot image recognition. Subsequent works, such as CoCoOp (Zhou et al., 2022a) and UPT (Zang et al., 2022), further conditioned textual prompts on visual features to enhance the model's generalization to unseen domains. Other methods, like KgCoOp (Yao et al., 2023), ProGrad (Zhu et al., 2023a), and ProReg (Zhu et al., 2023b), introduced prompt regularization strategies to minimize discrepancies between task-specific knowledge and the model's inherent general knowledge. Additional efforts explored prompt prior distribution (Lu et al., 2022), external knowledge (Shen et al., 2024), and optimal transport (Chen et al., 2022) for improving VLMs' performance on downstream classification tasks. Moreover, Maple (Khattak et al., 2023a) and PromptSRC (Khattak et al., 2023b) connected image and text prompts via linear projections to learn vision-textual prompts, while PromptKD (Zheng et al., 2024) introduced a prompt distillation framework to transfer knowledge from a large teacher model to a smaller student model.

While these methods have achieved substantial performance gains, their effectiveness significantly diminishes when test samples come from a different distribution. Our work focuses on test-time adaptation, enabling VLMs to adapt to test data without relying on the original training data.

**Test-Time Adaptation.** Test-time adaptation (TTA) (Sun et al., 2020; Zhang et al., 2022; Wang et al., 2020) aims to reduce distribution shifts between training and test data at test time, without accessing source data. Some approaches (Liu et al., 2021; Sun et al., 2020; Gandelsman et al., 2022) employ self-supervised proxy tasks to improve generalization, though these methods often require modifying the training process, limiting their applicability in the pre-trained model era.

To address this, fully test-time adaptation methods have been developed to adapt pre-trained models by enforcing self-consistency (Zhang et al., 2022), aligning train-test statistics (Mirza et al., 2022), or minimizing entropy in batch-wise predictions (Wang et al., 2020). Recent advances, such as TPT (Shu et al., 2022), extend entropy minimization to VLMs by learning textual prompts for each test sample. Other methods, like DiffTPT (Feng et al., 2023) and MTA (Zanella & Ben Ayed, 2024), leverage generated images from diffusion models (Rombach et al., 2022) to increase test data diversity. PromptAlign (Abdul Samadh et al., 2024) introduces distribution alignment by utilizing token distribution statistics, while TDA (Karmanov et al., 2024) designs dynamic adapters for efficient test-time adaptation in VLMs.

In contrast to these methods, our approach utilizes fine-grained attributes to guide models toward class-specific semantics, enabling better adaptation to the test domain through attribute selection and augmentation.

**Language-based Visual Recognition.** Visual recognition traditionally involves comparing an image to a fixed set of categories for classification. Recent advances have leveraged VLMs to achieve strong zero-shot recognition by utilizing both vision and language spaces. Early applications of VLMs relied on simple object names, often overlooking the rich contextual information (e.g., color, shape, texture) that language provides.

Some works (Zhai et al., 2024) enhance vision-language models by rewriting broad category descriptions using large language models (LLMs). Few-shot recognition methods (Yang et al., 2023b; Yan et al., 2023) decompose object recognition into attribute concepts, learning to classify images based on these attributes. Additionally, several prompt-tuning approaches (Tian et al., 2024; Mao et al.,

2023) improve model performance by utilizing attributes to refine the reasoning behind categorical predictions.

Unlike previous methods that use attributes to train shared networks or prompts for all test data, our approach Retrieves discriminative attributes tailored to each test sample, adapting VLMs for test-time visual recognition in a more precise and effective manner.

# 3 SEARCHING-FOR-PROMPTING FRAMEWORK

In this section, we first introduce a Retrieval-based Attribute Search (RAS) method to identify discriminative attributes for each test sample from attribute vocabularies. Subsequently, we describe the proposed Implicit-Explicit Attribute Injection (IEAI) technique, which effectively employs the searched attributes to enhance prompt learning in pre-trained models, as illustrated in Figure 3.

## 3.1 RETRIEVAL-BASED ATTRIBUTE SEARCH

**Baseline Model.** As a mainstream TTA method, test-time prompt adaptation Shu et al. (2022) provides the model with a context prompt tailored to each test sample, enabling it to recall the knowledge within CLIP more accurately. As labels are unavailable during test-time, a self-supervised optimization objective is often used for prompt tuning. Specifically, given a test sample $X_{\text{test}}$, the entropy of its prediction probability distribution is minimized to update the prompt $\boldsymbol{p}$ using the following objective function:

$$\mathcal{L}_{\text{self}} = \arg\min_{\boldsymbol{p}} -\sum_{i=1}^{C} \tilde{p}_{\boldsymbol{p}}(y_i|X_{\text{test}}) \log \tilde{p}_{\boldsymbol{p}}(y_i|X_{\text{test}}), \quad (1)$$

where $\tilde{p}_{\boldsymbol{p}}(y_i|X_{\text{test}})$ represent category probabilities produced by the model with learned prompt $\boldsymbol{p}$.

**Attribute Vocabulary.** Visual recognition tasks usually pre-defined category names and assign these categories to individual test samples during the test-time evaluation phase. Most existing TTA methods apply these pre-defined, coarse-grained category names in a straightforward way, thus leveraging only the limited knowledge embedded within the pre-trained models, especially in fine-grained recognition tasks that demand subtle discrimination. Large language models (LLMs) have showcased their ability to specify any visual concept using linguistic knowledge derived from an open-source vocabulary. To facilitate this, we systematically compile object categories from task-specific datasets to create an object vocabulary. Following Menon & Vondrick (2022), we produce detailed descriptions of the attribute features that uniquely characterize each object category in the vocabulary, by prompting LLMs with the input:

```
Q: What are useful features for distinguishing a {category
name} in a photo?
A: There are several useful visual features to tell there is
a {category name} in a photo: {attribute descriptions}.
```

By replacing {category name} with any object category, we can produce relevant {attribute descriptions} for the category vocabularies, thereby formulating the *attribute vocabularies* (see Figure 2).

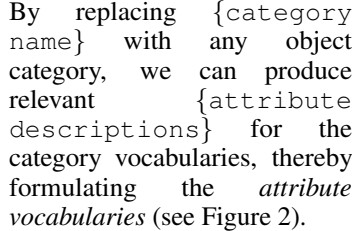
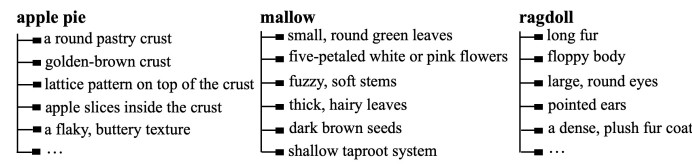

Figure 2: Examples of attribute vocabularies produced by GPT-3.

**Problem Definition.** We ground fine-grained attributes to their corresponding categories with text template, i.e., {category name} which (is/has/etc.) {attribute descriptions}. The reformulated attribute vocabularies $\mathcal{A} = \{a_1, a_2, \ldots, a_n\}$ offer substantial linguistic knowledge for vision recognition tasks; however, they also encompass considerable redundant information. *Consequently, the challenge for our test-time adaptation, where test data lacks labeled information, lies in how to retrieve discriminative attributes from $\mathcal{A}$ and use them to generate specific prompts for each test sample.*

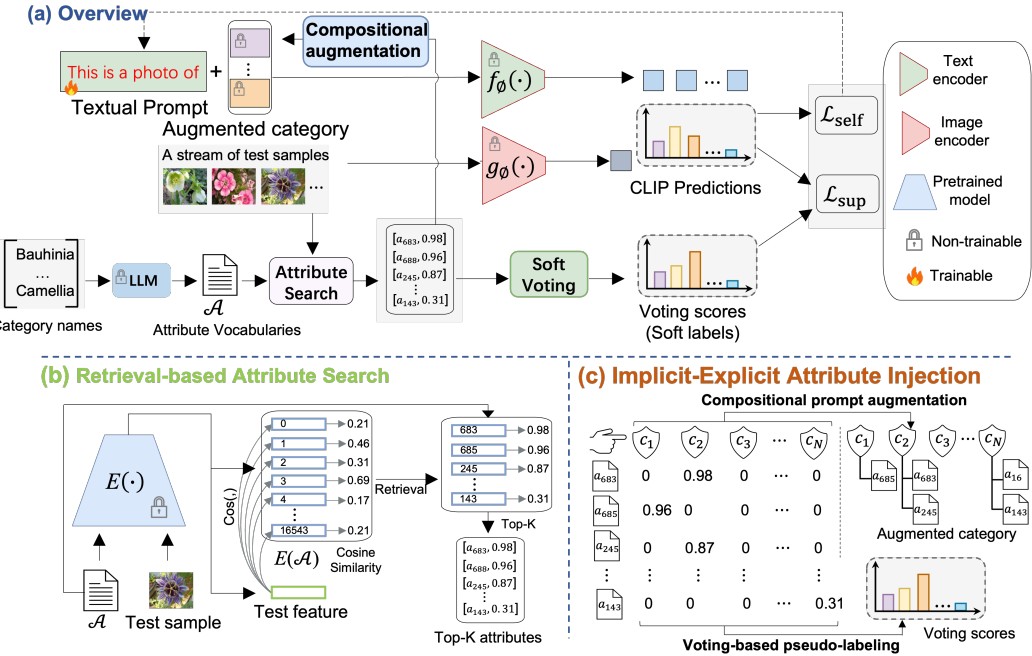

Figure 3: Architecture of our searching-for-prompting (i.e., Search4Prompt) framework. Given a test sample, our Search4Prompt retrieves the most discriminative attributes along with their relevance to predefined categories from attribute vocabularies. A soft voting mechanism is developed to leverage retrieval attributes for producing voting scores (i.e., soft labels) on predefined categories, which provides reliable labels for prompt learning with vision-language models. Our Search4Prompt also composites retrieved attributes to their corresponding categories, which further enhances the semantic representation of categories about the test samples and enables the model to use context information to discover better prompts.

**Discriminative Attribute Generation.** To address this challenge, we introduce RAS to filter out irrelevant attributes. Specifically, we initially construct an attribute feature bank by extracting textual features from the attribute vocabulary with auxiliary VLMs (e.g., CLIP Radford et al. (2021)). For each image, we utilize the auxiliary model to generate its visual feature $v$, which is then compared against the entries in the textual feature bank using cosine similarity. Subsequently, we identify the top-$K$ relevant attributes based on their alignment with attribute feature bank $\mathcal{A}$.

$$\hat{\mathcal{A}}_K = \arg \operatorname*{top-K}_{a \in \mathcal{A}} \cos(v, a), \tag{2}$$

where $\cos(\cdot, \cdot)$ indicates the cosine similarity and $\hat{\mathcal{A}} = \{\hat{a}_1, \hat{a}_2, \cdots, \hat{a}_K\}$ represents the top-$K$ relevant attributes for test sample.

**Discussion.** Our method aims to harness the capabilities of the existing foundational models to deliver comprehensive and accurate textual information for zero-shot vision recognition during test time. To achieve this, we utilize GPT-3 Mann et al. (2020) as the LLM to obtain attributes and the ViT-H-based CLIP Radford et al. (2021) as the auxiliary visual language model (VLM) to identify discriminative attributes. We also experiment with other auxiliary VLMs, such as ViT-B and ViT-G-based CLIP, to encode descriptive text. However, it is demonstrated that a smaller model has limited efficacy for fine-grained classification, as ViT-B-based CLIP tends to focus on more general attributes within the attribute vocabulary. Using Flower102 for instance, ViT-B-based CLIP tends to assign high similarity to broader attributes such as "Plant & Flower", whereas ViT-G-based CLIP recognizes more nuanced attributes like "white chest and paws & black markings on the face". Compared to existing methods that utilize only coarse-grained category names for test-time vision recognition, we explore the potential of incorporating fine-grained textual information to facilitate test-time prompt learning by using freely available public foundational models, which would be a new trend in the community. In addition, our approach has several advantages. Firstly, the RAS is an offline phase, where we only need to process all attribute vocabulary once, avoiding excessive computational overhead. Secondly,

with the generated descriptive attributes, we only need to train the vision recognition model. The auxiliary VLM can be substituted with other advanced foundational models, such as Florence Yuan et al. (2021) and Blip-2 Li et al. (2023). Consequently, the effectiveness of our method is likely to be enhanced in line with the advancements in these foundational models.

## 3.2 IMPLICIT-EXPLICIT ATTRIBUTE INJECTION

Although appending all the searched attribute descriptions behind learnable prompt vectors achieves more accurate predictions, the optimization of learnable prompts still lacks attribute-based guidance during test-time prompt learning. Thus, the learned prompts inevitably neglect the relationships across attributes contained in different categories, which leads to a sub-optimal solution, as shown in 1. To handle this drawback and fully utilize the searched attribute descriptions, we design an Implicit-Explicit Attribute Injection (IEAI), which mainly consists of a voting-based pseudo-label learning and a compositional augmentation strategy to enhance the model's zero-shot generalization ability.

### 3.2.1 VOTING-BASED PSEUDO-LABEL LEARNING

In light of the discriminative attribute description generated during the RAS phase, the primary challenge lies in effectively utilizing them to adapt VLMs to unlabeled test data. To address this, we propose a voting-for-prompting mechanism to softly vote attributes on predefined categories to generate voting scores that serve as pseudo-labels for test-time prompt learning. Concretely, this mechanism involves ranking attribute vocabulary, followed by voting within predefined task-specific categories to estimate the model predictions. Such predictions can guide model to learn textual prompts for different test samples using rich context information that language affords.

**Soft voting within object categories.** Traditional approaches often overlook the potential of soft labels derived from task-specific categories and their corresponding attributes. By designing a soft voting procedure within these categories, we aim to determine scores among top-$K$ attributes, which will serve as soft labels. These soft labels provide additional cues that guide the model to effectively utilize attribute information, enhancing its predictive capabilities through prompt learning. Specifically, given task-specific categories $\{c_1, c_2, \ldots, c_N\}$, we set the initial voting score for all categories to zero, and then traverse the top-$K$ grounding attributes for each category. If grounding attribute $a_k$ belongs to category $c_n$, we increase the voting score of the corresponding category by the matching value between them; otherwise, the score remains unchanged. Following this procedure, we derive the soft voting scores $\{s_1, s_2, \ldots, s_N\}$ for the test sample across the candidate categories, where higher scores reflect greater confidence in the model's predictions for the sample. To stabilize the training process, we apply softmax normalization to these voting scores,

$$p_i = \frac{e^{s_i}}{\sum_{j=1}^{N} e^{s_j}} \quad \text{for } i = 1, 2, \cdots, N. \tag{3}$$

Here $p_i, i = 1, 2, \cdots, N$, can be utilized as soft labels of the model to tune the textual prompt by minimizing cross-entropy loss,

$$\mathcal{L}_{\text{sup}} = -\sum_{i=1}^{K} p_i \log \tilde{p}_{\boldsymbol{p}}(y_i | X_{\text{test}}). \tag{4}$$

### 3.2.2 COMPOSITIONAL ATTRIBUTE AUGMENTATION

Existing methods typically enhance the discriminability among candidate categories using learned contextual prompts Shu et al. (2022) or rewriting language descriptions Fan et al. (2024). In our test-time adaptation paradigm, ambiguous categories vary across different test samples. To effectively mitigate category ambiguity tailored to the test sample, we design a customized attribute augmentation strategy to augment ambiguous categories with compositional attributes. Specifically, given a test sample from the downstream task, we determine candidate categories, which appear in the top-$K$ attribute vocabularies $\hat{\mathcal{A}}_K$, as its ambiguous categories. In our attribute vocabularies, each ambiguous category possesses multiple distinct attribute features. We composite these attributes into ambiguous categories to enhance their clarity and specificity. Compared to the original categories, the augmented categories exhibit stronger discriminative power through attributes, which enables the model to explore fine-grained attribute prompts and thus generalize better to test data.

## 4 EXPERIMENTS

### 4.1 EXPERIMENTAL SETUP

**Dataset.** We conducted main experiments on two benchmarks: the out-of-distribution (OOD) benchmark and the cross-domain benchmark. The OOD benchmark measures the robustness of our approach by assessing performance on four out-of-distribution datasets derived from ImageNet: ImageNet-A Hendrycks et al. (2021b), ImageNet-V2 Recht et al. (2019), ImageNet-R Hendrycks et al. (2021a), and ImageNet-S Wang et al. (2019). The cross-domain benchmark is utilized to evaluate the model's performance across ten diverse image classification datasets: Caltech-101 Fei-Fei et al. (2004), Oxford-Pets Parkhi et al. (2012), Stanford Cars Krause et al. (2013), Oxford-Flower102 Nilsback & Zisserman (2008), Food-101 Bossard et al. (2014), FGVC Aircraft Maji et al. (2013), EuroSAT Helber et al. (2019), SUN-397 Xiao et al. (2010), DTD Cimpoi et al. (2014), and UCF-101 Soomro et al. (2012).

**Implementation Details.** All models in our experiments are built upon the pre-trained CLIP model, which consists of an image encoder and a text encoder. By default, we employ a pre-trained CLIP model with a ViT-B/16 encoder as the backbone to ensure a fair comparison with existing methods Shu et al. (2022); Feng et al. (2023). Unless otherwise specified, we use GPT-3 to construct the attribute vocabularies and the CLIP ViT-H/14 backbone to extract their embeddings. Following Shu et al. (2022); Feng et al. (2023), the textual prompt is initialized in the hand-crafted default form, "a photo of a", and the corresponding four tokens are optimized based on a single test image. The prompt is optimized in three steps during the test phase using the AdamW optimizer with a learning rate of 0.001. The number of retrieved attributes $K$ is set as 15.

### 4.2 COMPARISONS WITH STATE-OF-THE-ARTS

In this section, we compare our proposed Search4Prompt with several state-of-the-art methods, including CLIP Radford et al. (2021), four train-time adaptation methods (i.e., CoOp Zhou et al. (2022b), CoCoOp Zhou et al. (2022a), MaPLe Khattak et al. (2023a), Prompt-SRC Khattak et al. (2023b)), as well as four existing test-time adaptation methods (i.e., TPT Shu et al. (2022), DiffTPT Feng et al. (2023), MTA Zanella & Ben Ayed (2024), AliYP Abdul Samadh et al. (2024), TDA Karmanov et al. (2024)). All train-time adaptation methods are

Table 1: Comparison with existing state-of-the-art methods on domain generalization evaluation. Green are trained on ImageNet using 16-shot training data per category and evaluated on cross-datasets. Blue are directly trained in an unsupervised manner on the test set and can dynamically adapt the model to each test sample.

| Method | I | A | R | V | K | Average |
|---|---|---|---|---|---|---|
| ViT-B/16 | 66.73 | 47.88 | 74.00 | 60.86 | 46.09 | 59.11 |
| CoOp | 71.51 | 49.71 | 75.21 | 64.20 | 47.99 | 61.72 |
| CoCoOp | 71.02 | 50.63 | 76.18 | 64.07 | 48.75 | 62.13 |
| MaPLe | 70.72 | 50.90 | 76.98 | 64.07 | 49.15 | 62.37 |
| PromptSRC | 71.27 | 50.9 | 77.80 | 64.35 | 49.55 | 62.77 |
| TPT | 68.98 | 54.77 | 77.06 | 63.45 | 47.94 | 62.44 |
| AliYP | - | 59.60 | 79.74 | **65.29** | 50.30 | 63.73 |
| DiffTPT | 70.30 | 55.68 | 75.00 | 65.10 | 46.80 | 63.16 |
| MTA | 69.29 | 57.41 | 78.33 | 63.61 | 48.58 | 65.01 |
| TDA | 69.51 | **60.11** | 80.24 | 64.67 | 50.54 | 65.01 |
| Ours | **70.54** | 57.91 | **81.01** | 64.19 | **56.79** | **66.09** |

tuned on ImageNet training data with 16-shot per class and tested on other datasets as in Shu et al. (2022). Different from train-time methods, test-time adaptation methods do not require training data and are tuned on testing datasets using a stream of unlabeled test samples. Following TPT Shu et al. (2022) and AliYP Abdul Samadh et al. (2024), we compare Search4Prompt with the state-of-the-art on domain generalization benchmark and cross-domain benchmark.

**Results on the Domain Generalization Benchmark.** We first compare Search4Prompt with state-of-the-art on domain generalization benchmark. Experimental results are summarized in Table 1. As shown in Table 1, the proposed Search4Prompt outperforms most existing methods across various OOD datasets from ImageNet. Specifically, our approach achieves average improvements of 3.32% and 1.08% over train-time and test-time adaptation methods, respectively. Even on in-domain dataset

Table 2: Comparison with existing state-of-the-art methods on cross-domain evaluation.

| Method | DTD | Flower | Pets | Cars | UCF101 | Caltech101 | Food101 | SUN397 | Aircraft | EuroSAT | Average |
|---|---|---|---|---|---|---|---|---|---|---|---|
| ViT-B/16 | 44.44 | 67.44 | 88.23 | 65.48 | 65.13 | 93.35 | 83.65 | 62.59 | 23.67 | 42.01 | 63.6 |
| CoOp | 41.92 | 68.71 | 89.14 | 64.51 | 66.55 | 93.7 | 85.30 | 64.15 | 18.47 | 46.39 | 63.88 |
| CoCoOp | 45.45 | 70.85 | 90.46 | 64.9 | 68.44 | 93.79 | 83.97 | 66.89 | 22.29 | 39.23 | 64.63 |
| MaPLe | 46.49 | 72.23 | 90.49 | 65.57 | 68.69 | 93.53 | 86.2 | 67.01 | 24.74 | 48.06 | 66.3 |
| PromptSRC | 46.87 | 70.25 | 90.25 | 65.70 | 68.75 | 93.60 | 86.15 | 67.10 | 23.90 | 45.50 | 65.81 |
| TPT | 47.75 | 68.98 | 87.79 | 66.87 | 68.04 | 94.16 | 84.67 | 65.5 | 24.78 | 42.44 | 65.1 |
| DiffTPT | 47.00 | 70.10 | 88.22 | 67.01 | 62.67 | 92.49 | 87.23 | 65.74 | 25.60 | 43.13 | 64.92 |
| AliYP | 47.24 | 72.39 | 90.76 | **68.50** | 69.47 | 94.01 | 86.65 | 67.54 | 24.80 | 47.86 | 66.92 |
| MTA | 45.59 | 68.26 | 88.22 | 68.05 | 68.11 | 94.13 | 84.95 | 64.98 | 25.32 | 38.71 | 64.63 |
| TDA | 47.40 | 71.42 | 88.63 | 67.28 | **70.66** | 94.24 | 86.14 | **67.62** | 23.91 | **58** | 67.53 |
| Ours | **60.17** | **75.44** | **93.00** | 65.03 | 68.17 | **95.05** | **87.73** | 65.79 | **25.74** | 54.02 | **73.82** |

Table 3: Ablation on the effectiveness of each design in our Search4Prompt approach. "Baseline" represents using entropy minimization on test samples; "Soft Voting" denotes the proposed voting-for-prompting mechanism; "Com. Aug." represents compositional attribute augmentation. Our proposed modules consistently outperform the baseline, and their combinations further enhance the model's recognition performance.

| Baseline | Soft Voting | Com. Aug. | DTD | Pets | Flower | A | R | Average |
|---|---|---|---|---|---|---|---|---|
| ✓ | | | 45.04 | 88.74 | 69.02 | 53.56 | 74.19 | 66.11 |
| | ✓ | | 54.31 | 90.79 | 74.95 | 55.44 | 78.79 | 70.86 |
| | | ✓ | 52.25 | 89.32 | 69.96 | 53.18 | 76.80 | 67.48 |
| ✓ | ✓ | | 57.39 | 92.55 | 75.40 | 56.45 | 81.01 | 72.56 |
| ✓ | | ✓ | 53.13 | 89.37 | 69.83 | 53.56 | 77.65 | 68.71 |
| | ✓ | ✓ | 57.8 | 92.50 | 75.13 | 55.94 | 79.74 | 72.22 |
| ✓ | ✓ | ✓ | **60.17** | **93.00** | **75.44** | **57.91** | **81.01** | **77.41** |

(i.e., ImageNet-I), our approach exhibits superior performance compared to existing test-time methods, while also exhibiting comparable performance to few-shot approaches. These results validate the effectiveness of Search4Prompt in enhancing test-time adaptation with the rich context information from attributes.

**Results on the Cross-domain Benchmark.** We further compare Search4Prompt with state-of-the-art methods on the cross-domain benchmark. Due to the wide distribution and varying granularity of these datasets, existing methods perform differently on different datasets. However, our method still achieves the best average performance, increasing the average accuracy from 67.53% to 73.82%. Specifically, our method provides consistent improvements on fine-grained datasets (e.g., Flower102 and Pets) and outperforms the previous best method. These results demonstrate that our approach can effectively recall the attribute knowledge contained within CLIP and transfer it to unseen domains in a zero-shot manner. However, it can be observed that our approach is inferior to the current best model on some coarse-grained datasets (e.g., UCF101 and SUN397). The main reason is that, on coarse-grained datasets, ambiguous category descriptions are relatively fewer compared to those in fine-grained datasets; thus, the model is able to easily recognize different objects with existing category descriptions.

### 4.3 ABLATION STUDY

**Components Ablation** In Tab. 3, we present an ablation study of the key components in Search4Prompt, including baseline, soft voting, and compositional augmentation (comp. aug.). The ablation experiments are conducted on five validation datasets, which include three fine-grained datasets (i.e., DTD, Pets, and Flower datasets) and two OOD datasets (i.e., ImageNet-A and ImageNet-R). As shown in Tab. 3, both soft voting and comp. aug. strategies achieve significant improvements over the baseline model, demonstrating that test-time adaptation can be improved by introducing attribute information with either pseudo-labeling or category description augmentation. Additionally, the two designs in Search4Prompt can complement each other as the combination of the two designs clearly outperforms either soft voting or comp. aug. on the benchmark datasets. Moreover, the two

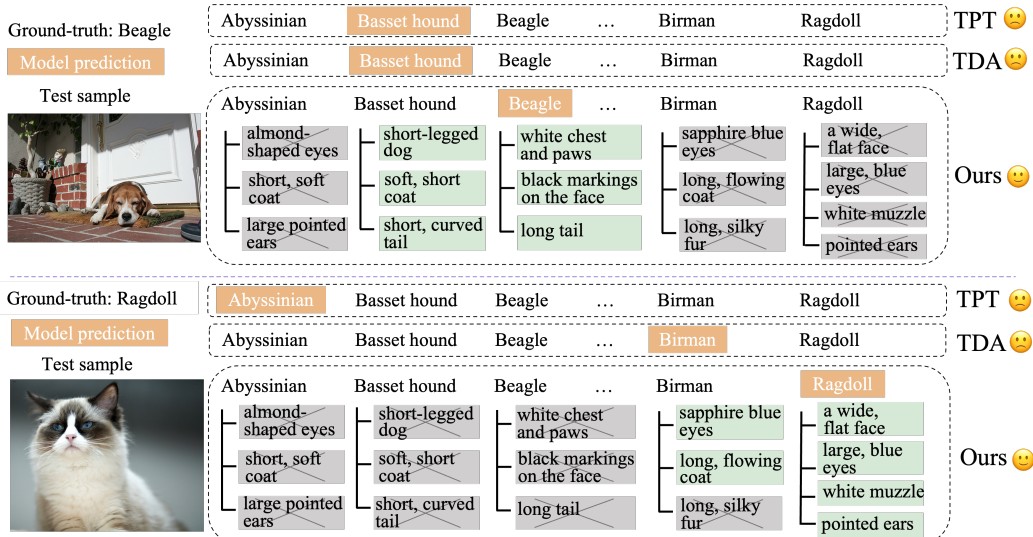

Figure 4: Qualitative comparison with existing methods. For each row, we show image examples from categories of DTD dataset. Existing methods often depend on broad category descriptions, which can result in incorrect predictions. Our method accurately captures the key attribute differences between categories and effectively utilizes them to enhance the model's zero-shot generalization ability.

designs are also compatible with the baseline model, and results show that their combinations (i.e., baseline+soft voting and baseline+comp. aug.) yield average improvements of 6.45% and 1.23%, respectively, over the baseline model. When all modules are employed concurrently, the model achieves the best performance, with an average improvement of 12.12% over the baseline. This indicates that soft voting and comp. aug. enable VLMs to capture fine-grained attribute information and enhance the semantic augmentation for ambiguous categories, while baseline model can transfer the fine-grained prior knowledge of pre-trained model to test data.

**The Superiority of Our Voting Mechanism During Soft Label Generation Process.** To demonstrate the superiority of our voting mechanism, we designed two variants: (i) Retrieval-based, which directly uses the retrieved attribute category with the highest similarity as a pseudo-label for the model, and (ii) Hard voting, which performs hard voting on predefined categories using the top-$K$ dis-

Table 4: Comparison with other soft label generation strategies on several benchmark datasets. Retrieval-based: directly using the retrieved attribute category with the highest similarity as soft label for the model. Hard voting: performing hard voting on predefined categories using the top-$K$ discriminative attributes.

| Method | DTD | Flower | Pets | A | R |
|---|---|---|---|---|---|
| Baseline | 44.44 | 67.44 | 88.23 | 47.88 | 74.00 |
| Retrieval-based | 57.86 | 70.77 | 92.40 | 52.63 | 77.58 |
| Hard voting | 59.04 | 73.24 | 92.70 | 55.61 | 79.12 |
| Soft voting (Ours) | **60.17** | **75.44** | **93.00** | **57.91** | **81.01** |

criminative attributes to produce pseudo-label. As shown in Tab. 4, our soft voting mechanism and its two variants surpass the baseline model. This improvement is attributed to the exploration of attribute information about the test sample to acquire a confident soft label. Furthermore, compared with the other two variants, our method achieves significant improvement over the baseline. The primary reason is that our method not only comprehensively considers different discriminative attribute categories, akin to hard voting, but also adapts the voting process based on their varying discriminative power.

**Comparison with VCD Menon & Vondrick (2022) in using LLM-generated descriptions.** VCD Menon & Vondrick (2022) also explores LLM-generated descriptions for categories. This work differs from our Search4Prompt as it naively encodes these descriptions and then compares them with image features. In Tab. 7, we compared our

approach with VCD Menon & Vondrick (2022), using the same text descriptions for a fair comparison. As shown in Tab. 7, VCD performs worse than our approach, indicating that VCD is not an optimal choice for utilizing LLM-generated descriptions. In fact, LLM-generated descriptions exhibit a degree of common sense but may not precisely align with specific samples used in downstream tasks. Our method facilitates the selection of the most representative attribute descriptions for various test samples, thereby adapting the model to the test data and enhancing performance.

Table 5: Comparison with VCD approach on benchmark datasets.

| Method | DTD | Flower | Pets | A | R |
|---|---|---|---|---|---|
| VCD | 42.43 | 69.23 | 81.30 | 38.87 | 54.41 |
| Ours | **60.17** | **75.44** | **93.00** | **57.91** | **81.01** |

**Qualitative Comparison with Existing Methods.** Figure 4 presents a qualitative comparison between our method and existing state-of-the-art methods (TPT and TDA) on the DTD dataset. Each row illustrates image examples from different categories. In the first row, the ground-truth label is "Beagle." Both TPT and TDA incorrectly predict the category as "Abyssinian" and "Basset hound," respectively. In contrast, our method accurately identifies the image as "Beagle." Similar results are observed in the second row. The comparable results show that our method can effectively captures discriminative attributes such as "white chest and paws" and "black markings on the face," which are distinctive to the Beagle category, while discarding other redundant attribute information. This demonstrates our method's superior ability to utilize detailed attribute information for accurate classification.

**The Number of Retrieved Attributes.** We conducted parameter studies on the number of retrieved attributes and reported the experimental results on the DTD and ImageNet-R datasets. Figure 5 illustrates that the performance of Search4Prompt is influenced when the number of attributes is either too low or too high. We found that setting the parameter to 15 for the DTD and ImageNet-R datasets yields the best performance. This is because an appropriate number of attributes ensures both representative and diverse attribute features in the implicit-explicit attribute injection process. Due to space limitations, we have moved additional experiments to the Appendix.

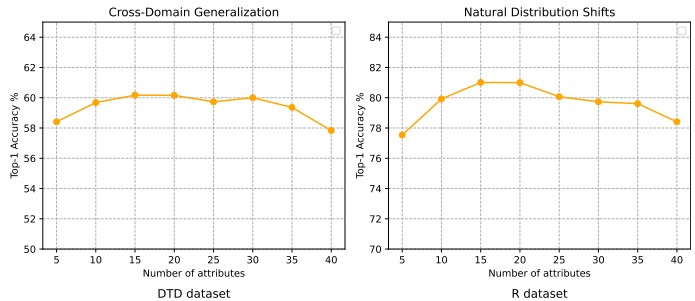

Figure 5: Parameter studies on the number of attributes under DTD and R datasets.

# 5 CONCLUSION AND LIMITATION

In this work, we investigated how to fully exploit the fine-grained attribute descriptions generated by LLM to enhance the zero-shot generalizability of vision-language models. We developed a Search4Prompt learning framework for test-time zero-shot recognition by leveraging visual attributes to guide the model toward class-specific modeling. The proposed Search4Prompt first searches discriminative attributes for test samples and then utilizes these attributes to learn tailored prompts that adapt VLMs to test data. We demonstrated the effectiveness of our method on the robustness to out-of-domain distribution shifts and cross-domain generalization.

**Limitation: Dependence on Visual Dictionary and Data.** The effectiveness of our method relies on a comprehensive attribute vocabulary. Despite the collection of numerous category names from existing datasets, there may still be deficiencies in specific tasks, such as Stanford Cars Krause et al. (2013) and FGVC Aircraft Maji et al. (2013). In such scenarios, our method may be unable to generate informative textual descriptions, leading to suboptimal results. Furthermore, in coarse-grained datasets, the existing category names are typically sufficient, and there are fewer ambiguous category descriptions. Consequently, our method may not exhibit its advantages in these datasets.

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

## 6 APPENDIX

Table 6: Evaluation on the auxiliary model

| Auxilliary Model | DTD | Flower102 | Pets | A | R | Average |
|---|---|---|---|---|---|---|
| ViT-B-16 | 46.04 | 68.29 | 88.47 | 48.61 | 74.07 | 65.10 |
| ViT-H-14 | 60.17 | 75.44 | 93.00 | 57.91 | 81.01 | 73.51 |
| ViT-bigG-14 | 60.5 | 76.61 | 94.28 | 58.13 | 82.45 | 74.40 |

**Auxiliary Model Evaluation.** In Tab. 6, we explore the effect of substituting the auxiliary model with ViT-B and ViT-G-based CLIP models. Experimental results demonstrate that using a more powerful model leads to superior performance. However, the average improvement becomes less significant when the model size exceeds that of ViT-H-based model.

**Comparison with VCD Menon & Vondrick (2022) in using LLM-generated descriptions.** VCD Menon & Vondrick (2022) also explores LLM-generated descriptions for categories. This work differs from our Search4Prompt as it naively encodes these descriptions and then compares them with image features. In Tab. 7, we compared our approach with VCD Menon & Vondrick (2022), using the same text descrip-

Table 7: Comparison with VCD approach on benchmark datasets.

| Method | DTD | Pets | A | R |
|---|---|---|---|---|
| VCD | 42.43 | 81.30 | 38.87 | 54.41 |
| Ours | **60.17** | **93.00** | **57.91** | **81.01** |

tions for a fair comparison. As shown in Tab. 7, VCD performs worse than our approach, indicating that VCD is not an optimal choice for utilizing LLM-generated descriptions. In fact, LLM-generated descriptions exhibit a degree of common sense but may not precisely align with specific samples used in downstream tasks. Our method facilitates the selection of the most representative attribute descriptions for various test samples, thereby adapting the model to the test data and enhancing performance.

Table 8: Comparisons of our Search4Prompt with TPT in terms of accuracy and testing time.

| Step | Method | DTD | | Flower102 | | Pets | | A | | R | |
|---|---|---|---|---|---|---|---|---|---|---|---|
| | | top-1 | time | top-1 | time | top-1 | time | top-1 | time | top-1 | time |
| 1 | TPT | 47.22 | 4.33 | 68.98 | 6.45 | 87.27 | 9.18 | 54.6 | 14.87 | 77.07 | 59.59 |
| | Ours | 56.86 | 5.04 | 73.97 | 7.27 | 92.67 | 7.49 | 56.32 | 30.70 | 78.27 | 119.36 |
| 3 | TPT | 47.34 | 5.25 | 68.45 | 9.10 | 87.14 | 10.95 | 58.39 | 36.06 | 77.34 | 143.26 |
| | Ours | 60.17 | 7.43 | 75.44 | 10.43 | 93.00 | 13.93 | 57.91 | 46.90 | 81.01 | 176.16 |
| 5 | TPT | 47.10 | 8.32 | 68.53 | 14.08 | 86.75 | 17.33 | 59.59 | 57.18 | 77.40 | 227.77 |
| | Ours | 61.05 | 9.83 | 75.36 | 13.61 | 93.05 | 15.34 | 57.95 | 59.40 | 80.54 | 235.85 |
| 7 | TPT | 47.16 | 11.16 | 68.57 | 19.11 | 86.86 | 23.09 | 59.96 | 77.83 | 77.35 | 310.32 |
| | Ours | 61.41 | 11.44 | 74.06 | 16.89 | 93.05 | 20.85 | 57.89 | 76.00 | 79.91 | 293.57 |

**Comparisons with TPT in terms of accuracy and testing time.** To provide a more comprehensive evaluation of Search4Prompt's efficiency and effectiveness, we compare it with the TPT method. This comparison encompasses both testing accuracy and time, with results obtained at different optimization steps. The evaluation is performed on five validation datasets, which include three fine-grained datasets (i.e., DTD, Pets, and Flower) and two OOD datasets (i.e., ImageNet-A and Imagenet-R). Both methods are executed on the same NVIDIA GPU server, and the experimental results are reported in Table 8. As shown in Table 8, when the optimization steps are minimal (i.e., 1), the proposed method demonstrates a significant improvement in testing accuracy across benchmark datasets, albeit with a sacrifice in testing efficiency. The time cost primarily arises from the necessity to search for fine-grained attributes during the inference stage to generate the model's fine-grained reward signals, thereby enhancing its generalization capability. As the number of optimization steps increases, our model's performance gradually improves. The performance will decrease slightly when the optimization continues beyond three steps. This indicates that additional optimization steps do not benefit the classifier; instead, a few updates are sufficient for the prompt to learn attribute information about the test samples.

