# OpenReview forum: "Test-time Zero-shot Recognition with Good Attributes"
_ICLR.cc/2025/Conference — ICLR 2025 Conference Withdrawn Submission_

### Official Review · Reviewer_pNAG · 2024-10-28

**Soundness:** 3
**Presentation:** 3
**Contribution:** 2
**Rating:** 3
**Confidence:** 4

**Summary:**

The paper presents a method for test-time zero-shot recognition that leverages attribute-based reasoning to improve model performance. The experiments are thorough, and the results are compelling.

**Strengths:**

The paper introduces a framework Search4Prompt to address the challenge of test-time adaptation in zero-shot learning scenarios. The two main components, Retrieval-based Attribute Search (RAS) and Implicit-Explicit Attribute Injection (IEAI) module contribute to the overall effectiveness of the framework. The paper provides extensive experimental results, demonstrating the effectiveness of Search4Prompt over existing methods on benchmark datasets.

**Weaknesses:**

1. The idea of searching relevant attributes for ZSL is not novel. For example, [a][b] show that it is possible to achieve the same recognition accuracy with a significantly smaller attribute vocabulary. What is the difference between these works? Besides, the proposed Retrieval-based Attribute Search is just a cosine similarity evaluation, which is somewhat simplistic.
2. The effectiveness of Search4Prompt relies on the comprehensiveness of the attribute bank. However, the details, such as how to determine the attribute bank and what attributes it concludes are not clear.
3. As we know, LLM is sensitive to prompts. The robustness of the framework to noisy or incorrect attribute descriptions from the LLMs is missing. It is also very important to evaluate the quality of generated attribute descriptions.

[a] Learning concise and descriptive attributes for visual recognition, ICCV'23.
[b] Language in a bottle: Language model guided concept bottlenecks for interpretable image classification, CVPR'23.

**Questions:**

See weaknesses above.

---

### Official Review · Reviewer_Gq2f · 2024-11-02

**Soundness:** 2
**Presentation:** 3
**Contribution:** 2
**Rating:** 5
**Confidence:** 3

**Summary:**

This paper presents a new test-time adaptation method that identifies critical attributes for prompt learning. The selected attributes are used in text prompt augmentation and pseudo labeling via an implicit-explicit attribute injection module. The experiments demonstrates the method's effectiveness.

**Strengths:**

1. The experimental results demonstrate the effectiveness of the proposed method.

2. This paper is well-written and easy to follow.

**Weaknesses:**

1.The proposed method reduces testing efficiency compared to the baseline, as the time required to search for attributes increases with the expansion of the attribute pool / number of categories, making it difficult to scale up. As shown in Table 8, the inference time considerably increases on ImageNet-R when using three optimization steps.

2.This paper lacks discussion and comparison with several prompt learning methods that also utilizes fine-grained attributes for VLM adaptation, i.e., MAP[1], ArGue[2].

[1] Argue: Attribute-guided prompt tuning for vision-language models. In CVPR, 2024.

[2] Multi-modal attribute prompting for vision-language models. TCSVT, 2024.

**Questions:**

1. It is weird that the few-shot setting and unsupervised setting are compared in the same table (Table 1). Can you provide results in few-shot learning experiments?

2. Could you offer a qualitative analysis of the generated and retrieved attributes for each category, with more examples?

3. Typos: Line 166 Retrieve.

---

### Official Review · Reviewer_Ucrt · 2024-11-03

**Soundness:** 2
**Presentation:** 3
**Contribution:** 2
**Rating:** 5
**Confidence:** 1

**Summary:**

This paper introduces a test-time adaptation method called Search4Prompt, which improves the TPT by identifying representative good attributes.

**Strengths:**

1. In general the paper is well organized and clearly written.

2. They retrieve and composite the discriminative attributes to tune prompts during test-time adaptation.

**Weaknesses:**

1. While the paper introduces an improved version of TPT by integrating text-based attributes, it would benefit from a clearer articulation of the specific innovations distinguishing it from existing TPT approaches. For instance, the authors could strengthen the novelty claim by further emphasizing unique aspects of the Retrieval-based Attribute Search (RAS) module or exploring novel attribute filtering techniques.

2. While RAS effectively retrieves attributes, it shows limitations in filtering irrelevant ones, especially when relying on LLM-generated attributes. To address this, it would be beneficial if the authors evaluated Search4Prompt with multiple LLMs to assess generalizability. Additionally, the paper could enhance its comparative analysis by including results from consistent auxiliary models, such as ViT-B-16-based CLIP, to provide a balanced perspective on performance variations under different model sizes (see Table 6).

3. It is not clear how the soft voting scores are derived from the top-k attributes. Following the approach described in Section 3.2.1, the class with the highest probability in the pseudo-labels in the IEAI in the Figure 1 should be "Red velvet cake," with the highest matching value of 0.23, rather than "cup cakes," which only reaches 0.22. The authors are encouraged to provide a step-by-step illustration or example of the soft voting process, explaining how these scores are calculated, as it would improve understanding and reproducibility.

4. Given the addition of 15 retrieved attributes, it would be valuable for the authors to provide a comparison of computational resources, such as memory usage, parameter count, and FLOPS, between their method and baselines (e.g., TPT and TDA). This addition would help readers assess the computational trade-offs associated with the approach.

5. The experimental results require further verification. First, the same baseline model yields different results in Table 3 and Table 4. Second, the VCD results in Table 5 appear to be questionable. For instance, VCD [1] achieves a zero-shot result of 86.92 on the Pets dataset without any prompts. Based on experience, a combination of VCD with the baseline TPT should yield improved results; however, this paper reports only 81.30. Similarly, in a related paper [2], CLIP + A evaluates pre-trained CLIP with attributes obtained from LLMs, achieving 80.84 on the Flower dataset, while this paper reports only 69.23. I recommend that the authors revisit and verify these results, providing further details on the experimental setup. Additionally, an analysis to reconcile the reported outcomes with those from related studies (e.g., VCD and CLIP + A) would enhance result validity and comparability.

[1] Menon, Sachit, and Carl Vondrick. "Visual classification via description from large language models." ICLR 2022.

[2] Saha, Oindrila, Grant Van Horn, and Subhransu Maji. "Improved Zero-Shot Classification by Adapting VLMs with Text Descriptions." CVPR 2024.

**Questions:**

No question

---

### Official Review · Reviewer_Jgu9 · 2024-11-03

**Soundness:** 3
**Presentation:** 3
**Contribution:** 3
**Rating:** 5
**Confidence:** 5

**Summary:**

This paper leverages descriptors generated by large language models (LLMs) to assist vision-language models (VLMs) at the inference stage, achieving performance improvements across multiple datasets.

**Strengths:**

1. The paper is well-written and organized, making it easy to follow.

2. Leveraging LLM-generated descriptors to enhance VLM performance during inference is a reasonable and meaningful approach.

3. The paper provides notable performance improvements over baseline methods, supporting the practical value of the approach

**Weaknesses:**

1. The approach of generating descriptors and the design of Discriminative Attribute Generation lack novelty; directly using top-k prototypes for each class at inference is also a common approach. Similar to Soft voting, DVDet [1] also selects high-confidence descriptors by voting, which can lead to ambiguous category selection through misclassification. Please clarify the distinctions.

2. CaF [2] should be used as a baseline to demonstrate the effectiveness of the method.

3. The statement, “The challenge for our test-time adaptation, where test data lacks labeled information, lies in how to retrieve discriminative attributes from A and use them to generate specific prompts for each test sample,” is unnecessary, as test data inherently lacks label information.

References:

[1] LLMS MEET VLMS: BOOST OPEN VOCABULARY OBJECT DETECTION WITH FINE-GRAINED DESCRIPTORS
[2] Visual Classification via Description from Large Language Models

**Questions:**

Please see the weakness.

---

### Note · Authors · 2024-11-15

I have read and agree with the venue's withdrawal policy on behalf of myself and my co-authors.